# Spatial Structuring of Soil Fungal Diversity Associated with *Ziziphus lotus* (Rhamnaceae) in Arid Agricultural Soils

**DOI:** 10.3390/microorganisms13112489

**Published:** 2025-10-30

**Authors:** Nabil Radouane, Salma Mouhib, Khadija Ait Si Mhand, Zakaria Meliane, Khaoula Errafii, Mohamed Hijri

**Affiliations:** 1African Genome Center, University Mohammed VI Polytechnic (UM6P), Lot 660, Hay Moulay Rachid, Ben Guerir 43150, Morocco; nabil.radouane-ext@um6p.ma (N.R.); salma.mouhib@um6p.ma (S.M.); khadija.aitsimhand@um6p.ma (K.A.S.M.); zakaria.meliane@usherbrooke.ca (Z.M.); khaoula.errafii@um6p.ma (K.E.); 2Institut de Recherche en Biologie Végétale, Département de Sciences Biologiques, Université de Montréal, 4101 Rue Sherbrooke Est., Montréal, QC H1X 2B2, Canada

**Keywords:** arid ecosystems, community structure, diversity, fungi, spatial distribution, soil, *Ziziphus lotus*

## Abstract

*Ziziphus lotus* (L.) Lam., (Rhamnaceae) a resilient shrub native to Moroccan’s arid regions, functions as a keystone species by creating microhabitats that buffer temperature extremes, retain soil moisture, and accumulate organic matter. However, its role in structuring soil fungal diversity and community composition in these environments remains largely unexplored. This study investigated the spatial distribution of fungal communities associated with *Z. lotus* in barley-planted and non-planted fields. Soil samples were collected at 0, 3, and 6 m from shrub clusters during the barley harvest. The fungal community was dominated by Ascomycota (93.5%). Alpha diversity indices (Shannon–Wiener and Simpson) were significantly higher near shrub bases (0 and 3 m) compared to more distant soils (6 m), indicating a clear decline in diversity with distance (0 m vs. 6 m: *p* = 0.0012; 3 m vs. 6 m: *p* = 0.0007). Soil physicochemical parameters, including calcium carbonate content, nitrate, and salinity, significantly influenced fungal diversity (*p* ≤ 0.05). Beta diversity analysis revealed significant spatial differentiation in fungal community composition (PERMANOVA: *p* = 0.001). Overall, fungal richness and diversity were highest near shrub. Genera such as *Cladosporium*, *Fusarium*, and *Alternaria* were more abundant near shrub bases, while taxa like Didymellaceae and *Alfaria* were specially restricted. Functional predictions indicated dominance of fungi with mixed trophic modes (pathotroph–saprotroph–symbiotroph), suggesting ecological plasticity. Despite barley cultivation, the fungal community structure remained largely similar between the planted and non-planted fields. Overall, our findings underscore the ecological importance of *Z. lotus* as a reservoir of stress-tolerant fungi and as a potential keystone species for restoring degraded arid ecosystems.

## 1. Introduction

*Ziziphus lotus* (L.) Lam. (Rhamnaceae) commonly known as wild jujube, is a perennial shrub widely distributed across arid and semi-arid regions of North Africa, where it plays a key ecological role [1]. Renowned for resilience, *Z. lotus* withstand harsh abiotic conditions, including prolonged drought, high temperatures, and poor soil fertility [2], as well as biotic stressors such as parasitism by *Cuscuta epithymum* (L.) L. [3]. As nurse plant, it contributes to ecological stability by forming dense vegetation clusters that provide shade, lower surface temperatures, and retain soil moisture [4,5]. These microhabitats support the establishment of other plant species and offer refuge to insects, birds, reptiles, and small mammals [6]. Furthermore, the shrub’s deep root system stabilizes soil, mitigates erosion, and promotes nutrient cycling [7], thereby contributing to the formation of fertile “resource islands” [8]. Due to these functions, *Z. lotus* is considered a potential keystone species for the conservation and restoration of degraded drylands [9].

Like many desert plants, *Z. lotus* hosts a diverse fungal community that includes foliar endophytes, root-associated fungi, and rhizosphere inhabitants. Nearly all plants in natural ecosystems form associations with mycorrhizal or endophytic fungi, which enhance their adaptability under environmental stress [10]. Several studies have demonstrated the rich fungal diversity associated with *Ziziphus* species. For example, a survey in Oman reported over 80 endophytic fungal species from healthy *Ziziphus* leaves, including *Alternaria* spp., *Aspergillus* spp., *Cladosporium* spp., and *Fusarium* spp. [11]. Similar findings from North Africa confirmed the predominance of Ascomycota (~78%), followed by Basidiomycota, suggesting a conserved foliar mycobiota across regions [12].

These fungi often engage in mutualistic interactions enhancing host fitness through stress mitigation, growth promotion, and production of bioactive metabolites [12]. In particular, *Z. lotus* is known to form arbuscular mycorrhizal (AM) associations with glomeromycetes, which aid in water and nutrient uptake, especially phosphorus, crucial for survival in nutrient-depleted soils [13]. Inoculation studies with *Rhizophagus irregularis* have demonstrated a ~70% increase in biomass and phosphorus uptake in *Ziziphus* under AM colonization [14], underscoring the functional relevance of its fungal symbionts.

Despite these benefits, *Z. lotus* poses challenges in agroecosystems. Its extensive root system competes with crops for water and nutrients, potentially reducing yields and complicating land management [15]. Its thorny branches can damage agricultural machinery and hinder field operations. Therefore, while conservation is essential in natural habitats, integrated management strategies are required in agricultural contexts to balance ecological functions with productivity.

The role of *Z. lotus* in structuring soil fungal diversity in agricultural landscapes remains poorly understood. This study investigates whether wild jujube shrubs harbor fungal communities of ecological relevance in arid agroecosystems and whether these communities exhibit spatial structuring across shrub gradients. Given the well-known fertile island effect of desert shrubs, characterized by the accumulation of organic matter and improved microclimate beneath the shrubs, we expect fungal diversity to be highest under or near the *Z. lotus* canopy and to decline with increasing distance into the open field. Our objectives were (i) to assess spatial patterns of fungal diversity across *Z. lotus* shrub gradients; (ii) to identify the dominant fungal taxa and their potential ecological functions under both cultivated and fallow conditions; and (iii) to evaluate differences in fungal community composition between cultivated (planted) and fallow (non-planted) fields.

## 2. Materials and Methods

### 2.1. Study Site and Sample Collection

Sampling was conducted on 16 May 2023, in Rhamna Province, Morocco (GPS coordinates: 31°59′44″ N, 8°01′02″ W), approximately 100 m from Highway A3 [16]. The study site comprised two adjacent 1-hectare fields: one actively planted with barley and occasionally irrigated during the growing season, and the other left non-planted (Appendix A).

The Rhamna region lies at the interface between Mediterranean and Saharan climatic zones, exhibiting a semi-arid to arid climate. Summers are long, hot and dry, with daytime temperatures exceeding 40 °C in July and August. Winters are short and mild, with average temperatures ranging from 10 to 15 °C, and occasional nighttime lows below 5 °C. Annual precipitation ranges from 200 to 400 mm, concentrated primarily between November and March, with negligible rainfall during summer months [17].

To investigate the influence of *Z. lotus* on soil fungal communities, a spatially explicit sampling design was implemented. Five shrub patches were sampled within the planted field and two in the adjacent non-planted field. Each patch was sampled at 0 m (beneath the shrub canopy), 3 m, and 6 m representing a distances gradient from immediate shrub influence to surrounding field soil. These distances were chosen because the average spacing between distinct shrub patches was ~12 m; thus 6 m represents roughly half the inter-patch distance (minimizing overlap between neighboring shrubs’ influence), and 3 m serves as an intermediate point just outside the canopy. Each sample was a composite of four subsamples (~250 g each), collected at a depth of 20 cm (Appendix A). These distances were chosen based on the minimum distance (12 m) between shrub patches. In total, 84 soil samples were collected: 60 from the planted field and 24 from the non-planted field. Samples were sealed in Ziplock plastic bags (30 × 20 cm), stored on ice, and transported to the laboratory for analysis.

### 2.2. Soil Physicochemical Analysis

Composite samples representing each distance class (0 m, 3 m, and 6 m) within each *Z. lotus* patch were homogenized and analyzed at the AITTC laboratory, University Mohammed VI Polytechnic (Benguerir, Morocco). The analysis included measurements of physical properties of soil (Clay, Silt and Sand) using the NF X 31-107 protocol [18]. Soil pH was measured according to NF ISO 10390 [18], while electrical conductivity (EC) was determined following NF ISO 1 [18]. Available nutrients (nitrate-N, P_2_O_5_, K_2_O) were analyzed using NF ISO 11263 [18]. Micronutrients (Fe, Zn), cation exchange capacity (CEC), organic matter (OM), calcium carbonate (CaCO_3_), sodium oxide (Na_2_O), magnesium oxide (MgO), and calcium oxide (CaO) were determined according to NF X 31-108, NF X 31-121, and NF X 31-130, respectively, following standard soil analysis protocols (Appendix A) [16].

### 2.3. DNA Extraction, PCR Amplification, and Sequencing

Genomic DNA was extracted from 250 mg of soil from each of the 84 samples using the Soil Pro Kit (Qiagen; distributed by Global Diagnostic Distribution, Témara, Morocco), following the manufacturer’s instructions. Samples were homogenized using a TissueLyser II (Qiagen) with 2 mm tungsten beads at 24 Hz for 15 min. DNA integrity was assessed via agarose gel electrophoresis, and concentration was measured with a BioSpectrophotometer (Eppendorf, Hamburg, Germany). Fungal ITS2 region amplification was performed using the primers CS1_ITS3_KYO2 (5′-ACACTGACGACATGGTTCTACAGATGAAGAACGYAGYRAA-3′) and CS2_ITS4 (5′-TACGGTAGCAGAGACTTGGTCTTCCTCCGCTTATTGATATGC-3′) (Alpha DNA, Montreal, Canada) as described in Legeay et al. [19]. Each 25 μL PCR reaction contained 1× Platinum Direct PCR Universal Master Mix (ThermoFisher, Rabat, Morocco), 0.25 μM of each primer, and ~10 ng of DNA template. Amplifications were performed in duplicate on a Mastercycler X50s (Eppendorf) using the following thermal profile: initial denaturation at 94 °C for 3 min; 30 cycles of 94 °C for 30 s, 55 °C for 30 s, and 72 °C for 1 min; final elongation at 72 °C for 7 min. Negative (sterile water) and positive controls were included.

The fungal ITS amplicon library preparation was performed as described in Legeay et al. [19]. Briefly, the PCR products were purified using Agencourt AMPure XP beads (Beckman Coulter, Brea, CA, USA). After two ethanol washes and air drying, amplicons were resuspended in 10 mM Tris (pH 8.5). A second PCR was conducted to add Illumina sequencing adapters and index tags, using 5 µL purified PCR product, 2.5 µL Fluidigm Access Array Barcode 384, and 1× KAPA HiFi HotStart ReadyMix (Roche Sequencing Solutions, Santa Clara, CA, USA). Thermocycling was as follows: 95 °C for 3 min; 8 cycles of 95 °C for 30 s, 55 °C for 30 s, 72 °C for 30 s; and a final extension at 72 °C for 5 min.

Final libraries were purified with AMPure XP beads, quantified using the Qubit dsDNA HS Assay Kit (ThermoFisher, Témara, Morocco), and normalized and pooled according to Illumina protocols. Paired-end sequencing (2 × 300 bp) was performed using a MiSeq V3 reagent kit on an Illumina MiSeq platform (Illumina, Paris, France).

### 2.4. Bioinformatic Analysis

Raw sequence data was processed using the DADA2 pipeline in R v4.3.3 [20]. Low reads (Q < 30) were discarded. Primers and adapters were trimmed and reads denoised based on DADA2’s error model amplicon sequence variants (ASVs). Taxonomic assignment of ASVs was performed using BLAST against the UNITE database [21], via *assignTaxonomy* function.

### 2.5. Statistical Analysis

ASVs abundances were transformed into compositional data using the *transform* function of the *phyloseq* package (v1.46.0) with the “compositional” option to normalize sequencing depth.

Alpha diversity was calculated using Shannon–Wiener, inverse-Simpson, Pielou’s evenness, Simpson evenness, observed richness, Chao1, and ACE indices with the *phyloseq* and *vegan* packages (v2.6-8) [22]. Beta diversity was assessed using Bray–Curtis dissimilarity matrices and tested for significance using PERMANOVA (*adonis* function, *vegan* package) [23]. Soil variables were analyzed using univariate and multivariate ANOVAs to assess spatial trends across distance gradients. We also performed one-way ANOVA to compare diversity across the three distances, followed by post hoc tests to identify pairwise differences. Network analyses were conducted using *SpiecEasi* (v1.1.2) [24] and *igraph* (v2.0.2) axon-level betweenness centrality was used to identify potential hub taxa mediating community interactions.

Random forest models were built using the *randomForest* package (v4.7-1.2) [25] with 100 trees per model to classify fungal community patterns across conditions. The purpose of this model was to classify fungal community composition across different conditions (in this case, distances from the shrub), and to identify the taxa most important for distinguishing among those conditions. Taxon importance was evaluated using the model’s importance scores, specifically the Mean Decrease in Accuracy metric. The most informative taxa were identified based on model importance scores. Indicator species analysis was performed using the *indicspecies* package (v1.9.0) [26] to identify taxa significantly associated with specific environmental conditions or sampling distances.

## 3. Results

### 3.1. Soil Physicochemical Composition

Multivariate analysis revealed several soil variables as significant predictors of alpha diversity (Appendix A). For the Shannon–Wiener index, calcium carbonate (CaCO_3_), sodium oxide (Na_2_O), and nitrate (NO_3_^−^) were significant (*p* ≤ 0.05), with Na_2_O showing the strongest influence (*p* = 0.001 ***), suggesting that sodium oxide has a substantial impact on fungal diversity. The same three variables were also significant for the Simpson index, underscoring their consistent influence on community structure.

In univariate analyses (Appendix A), organic matter, Na_2_O, and total nitrogen (Total N) were significantly correlated with both Shannon–Wiener and Simpson indices (*p* ≤ 0.05), highlighting the individual contributions of nutrient content and sodium concentration. Magnesium oxide (MgO) exhibited particularly strong effects (*p* ≤ 0.001 for Shannon–Wiener; *p* < 0.001 for Simpson), indicating magnesium’s central role in shaping diversity. Additionally, calcium oxide (CaO) was significant for the Simpson index (*p* = 0.045), suggesting a potential influence on community evenness and dominance.

PERMANOVA (adonis2) analysis for Total P, Total N, and Total K (Table 1) showed that at 0 m, these variables significantly shaped community structure (*p* = 0.029, 0.004, and 0.024, respectively). However, no significant effects were observed at 3 m (*p* = 0.547–0.784), and at 6 m, while *p*-values were lower (0.195–0.506), none reached statistical significance. These results suggest that nutrient-driven structuring of microbial communities is strongest at the plant origin point, diminishing with distance.

### 3.2. Fungal Diversity and Richness Vary with Special Distribution

Sequencing yielded 282,652 reads, which were clustered into fungal ASVs (Appendix A), spanning 204 genera, 107 families, 52 orders, 22 classes, and 7 phyla. ASV counts varied by distance: 774 ASV at 0 m, 978 at 3 m, and 579 at 6 m. Uniquely, 28.6% of ASVs were exclusive to 0 m, and 39.9% were unique to 3 m. Ascomycota dominated across all samples (93.5% of ASVs), followed by Basidiomycota (5.2%) and Mortierellomycota (0.6%). The most abundant genera overall were, *Alternaria* (23.7%), *Ascobolus* (11.4%), and *Fusarium* (10.5%) (Figure 1).

At the individual sampling distances, the composition of fungal phyla and dominant genera showed notable variation. At 0 m, the community was overwhelmingly dominated by Ascomycota (96.3%), followed by Basidiomycota (2.4%) and Mortierellomycota (1.0%). The most abundant genera at this distance were *Alternaria* (21.9%), *Ascobolus* (10.1%), and *Fusarium* (9.2%). At 3 m, Ascomycota remained dominant but decreased slightly to 92.0%, while Basidiomycota increased to 6.3%, and Mucoromycota appeared at 0.9%. The leading genera were *Alternaria* (23.8%), *Ascobolus* (12.5%), and *Fusarium* (9.9%). At 6 m, the relative abundance of Ascomycota was similar (92.1%), but Basidiomycota increased further to 7.3%, and Mucoromycota declined to 0.5%. The top genera at this distance shifted to *Alternaria* (26.3%), *Fusarium* (13.4%), and *Cladosporium* (13.1%), suggesting a compositional shift in fungal communities with increasing distance from the plant. Beyond the effect of distance, overall fungal diversity was higher in the barley-planted field compared to the non-planted field. Shannon–Wiener diversity in planted soil was significantly greater (*p* = 0.0051) than in non-planted soil, and other indices, such as richness and evenness, also showed higher values in the planted field (*p* ≈ 0.01–0.04).

PERMANOVA on Bray–Curtis distances confirmed significant differences in community composition across distances and field conditions (*p* = 0.001; Table 2), through the interaction between distance and field distance was marginal (*p* = 0.056). Alpha diversity indices further highlighted spatial trends. The Shannon–Wiener index did not differ significantly between the 0 m and 3 m (*p* = 0.747) but was significantly higher in both compared to 6 m (*p* = 0.001 and *p* = 0.0007, respectively (Table 3)). Richness indices, including Richness (ACE, and Chao1, and observed richness) followed the same trend, with 6 m showing markedly lower diversity (*p* ≤ 0.001). In contrast, evenness indices such (Pielou’s and Simpson) did not vary significantly by distance. Planted plots exhibited significantly higher diversity than non-planted ones in terms of Shannon–Wiener (*p* = 0.0051), richness (*p* = 0.034), Pielou evenness (*p* = 0.012), and ACE (*p* = 0.039), while no differences were found for Simpson evenness or Chao1. The contrast between 3 m and 6 m was particularly pronounced (*p* = 0.001624), while 0 m vs. 3 m remained non-significant. These trends are visualized in Figure 2, where only comparisons involving 6 m revealed significant differences.

### 3.3. Fungal Communities Are Structurally Shaped by Distance

Beta diversity analysis confirmed a clear special distribution of fungal communities. PCoA based on Bray–Curtis dissimilarities (Figure 3) revealed loose clustering at 0 m, while 3 m and 6 m samples were more widely scattered. An ADONIS test confirmed significant effects of distance on community composition (*p* = 0.002). Although the effect size was modest, these results indicate that agricultural cultivation exerts a detectable influence on community composition. Planted fields tended to cluster separately from non-planted field, reflecting differences in the relative abundances of several taxa. Nonetheless, the substantial overlap between groups suggests the presence of a shared core fungal community across both land-use types.

### 3.4. A Shared Core Coexists with Distance-Specific Fungal Assemblages

At the genus level (Figure 4A), 3 m samples harbored the most unique genera (37, 18.1%), followed by 0 m (34, 16.7%) and 6 m (8, 3.9%). 72 genera (35.3%) were shared across all distances. At the ASV level (Figure 4B), 3 m again showed the highest number of unique ASVs (637, 35.5%), compared to 467 (26.0%) at 0 m and 304 (16.9%) at 6 m. Only 149 ASVs (8.3%) were common for all distances.

Random forest classification identified the top 20 ASVs most responsible for differentiating distances (Appendix A, Figure 5). ASV_96 (Dothidotthiaceae, unassigned genus) ranked highest in importance (MeanDecreaseAccuracy = 10.55). Other key taxa included ASV_8 (*Pleiochaeta*), ASV_93 (*Aureobasidium*), and ASV_62 (Chaetomiaceae), pointing to the roles of Dothideomycetes and Sordariomycetes in spatial differentiation.

Multiple ASVs from Pleosporales (ASV_106, ASV_122, ASV_24) and *Fusarium*-related taxa (ASV_60) were important markers, highlighting the sensitivity of Hypocreales to spatial gradients. ASVs from Saccotheciaceae (ASV_30, ASV_34) also differentiated near-origin from distal sites.

Figure 5 visualizes these associations, showing taxa such as ASV_8 (*Pleiochaeta*) and ASV_5 (unassigned) as strongly responsive to distance. Genera including *Xenodidymella*, *Alternaria*, and *Cladosporium* contributed moderately, suggesting widespread but spatially variable presence.

### 3.5. Functional Potential Varies with Spatial and Planting Conditions

FUNGuild functional prediction (Top 20 taxa shown) revealed that most fungi exhibited mixed trophic modes, primarily pathotroph-saprotroph or pathotroph-saprotroph-symbiotroph, indicating potential roles as decomposers, pathogens, and symbionts. At the near-shrub (0 m), multitrophic taxa were most prevalent, including *Pleiochaeta*, *Xenodidymella*, *Aureobasidium*, and *Cladosporium* (pathotroph–saprotroph [±symbiotroph] assignments), consistent with a resource-rich microhabitat favoring ecological generalists. Typical fast-growing saprotrophs such as *Chaetomium* and *Penicillium* were also enriched under the canopy, reflecting litter/OM inputs and rapid turnover. By contrast, at 3–6 m, we observed higher relative representation of oligotrophy-tolerant taxa such as *Alfaria* and *Myriococcum*, consistent with lower nutrient availability away from shrubs. Across land-use types, the cultivated field was enriched in Cystofilobasidiaceae (yeast-like taxa often classified as saprotrophs or mixed modes), whereas Stephanosporaceae, *Myriococcum*, and *Alfaria* were relatively more abundant in the fallow field (Figure 6).

Differential abundance analysis (Appendix A) revealed significant compositional shifts between planted and non-planted fields. Several taxa, including Cystofilobasidiaceae, *Filobasidium*, *Stempellium*, and *Cladosporium*, were significantly enriched under planted conditions, suggesting that crop presence and associated root inputs promote these fungal groups. In contrast, taxa such as *Stephanosporaceae*, *Myriococcum*, and *Alfaria* were more abundant in non-planted field, reflecting adaptation to resource-poor environments or ecological specialization in the absence of cultivation. These contrasting patterns indicate that plant presence shapes fungal community assembly by favoring distinct ecological strategies.

Co-occurrence patterns (Appendix A) reinforced these trends. In planted fields, ASV_8 (*Pleiochaeta*) was a dominant member, potentially reflecting adaptation to the rhizosphere. In non-planted fields, ASV_57 (*Aspergillus*) prevailed, consistent with its saprophytic lifestyle in plant-free soils. These patterns suggest that fungal taxa differentially respond to the presence or absence of vegetation, likely due to changes in root exudates, soil nutrients, or organic inputs.

## 4. Discussion

This study investigated the spatial distribution of fungal communities surrounding wild jujube shrubs in arid environment, highlighting their potential ecological roles under contrasting land uses. The results revealed a heterogeneous fungal community structure, with clear spatial patterns in both cultivated and uncultivated fields. Together with the findings of Radouane et al. [16], who reported spatial structuring of bacterial communities around *Z. lotus* patches and demonstrated the shrub’s substantial influence on bacterial dynamics in arid ecosystems, our results support the hypothesis that fungal richness and composition vary with distance from *Z. lotus* and that the shrub exerts ecological effects extending beyond its immediate root zone.

As a keystone species in arid ecosystems, *Z. lotus* creates localized microhabitats that enhance fungal diversity compared to surrounding barren soils. By accumulating litter and nutrients under its canopy, the shrub promotes microbial activity and alters soil conditions in ways that support diverse fungal communities [27]. Similar patterns of higher fungal richness under shrub canopies compared to open soils have been reported for other desert shrubs [28]. This effect is likely mediated through organic matter inputs, shade, and moisture retention, which allow a broader range of fungal taxa to persist.

The fungal communities associated with *Z. lotus* displayed both spatial and functional differentiation, shaped by resource gradients and land use. Our findings confirmed elevated levels of organic matter, nitrogen (N), and phosphorus (P) at 0 m compared to farther distances, validating this shrub-mediated nutrient enrichment. Notably, the abundance of *Alternaria*, *Fusarium*, and *Cladosporium* near roots (0 m), and the exclusive presence of Didymellaceae at 0 m and *Alfaria* at 6 m, point to niche specialization influenced by root exudates and local organic matter availability. Functional predictions from FUNGuild further demonstrated that many dominant taxa exhibit mixed trophic strategies, suggesting high ecological plasticity, a necessary trait in environments where nutrients and water are limiting [29].

The dominance of Ascomycota across all distances is consistent with previous studies in arid regions [28,30]. This phylum is well known for its tolerance to extreme stressors, including drought, salinity, and high temperatures, owing to adaptations such as melanized cell walls and stress-responsive metabolisms [31,32]. Our data also revealed the presence of unclassified fungal taxa (e.g., ASV_115: *Ascobolaceae* incertae sedis), reinforcing the idea that *Z. lotus* hosts underexplored fungal lineages potentially adapted to harsh conditions.

Specific classes within Ascomycota, Dothideomycetes, Sordariomycetes, and Pezizomycetes, emerged as spatial indictors. Genera like *Pleiochaeta*, *Neomicrosphaeropsis*, and *Alternaria* showed clear spatial trends, with intra-generic variability (e.g., *Fusarium*) contributing to spatial differentiation. The detection of *Rhizophlyctis* (Chytridiomycota) also suggests that, despite Ascomycota dominance, other fungal lineages contribute to community structuring. These patterns reflect fungal sensitivity to fine-scale environmental gradients and suggest that certain taxa can serve as bioindicators of spatial habitat variation.

In this study, our results demonstrate that *Z. lotus* strongly influences the composition and diversity of soil fungal communities through localized environmental modifications. Nutrient enrichment beneath shrub canopies provides resources that support a richer fungal assemblage. In addition, the shrub canopy likely buffers soil microclimate by moderating temperature and conserving moisture, conditions known to favor microbial persistence in arid soils. Root exudates and potential mycorrhizal associations may further select for specific taxa, such as *Pleiochaeta* and *Aureobasidium*, that thrive in rhizosphere-enriched niches. While fungi are well documented to enhance plant stress tolerance in other systems, our study did not measure plant performance, so we cannot conclude a direct benefit to *Z. lotus*. Instead, our findings highlight the intrinsic mechanisms by which shrubs shape soil fungal communities, creating fertile islands that sustain microbial diversity under harsh environmental conditions. Fungal partners can facilitate nutrient acquisition and improve drought tolerance [33].

Alpha diversity was highest in soils immediately beneath *Z. lotus,* decreasing with distance, a pattern consistent with the “fertile-island” effect, where shrubs enrich microbial life beneath canopies [28]. Conversely, soils farther from the shrub were more exposed and nutrient-poor, supporting less diverse communities. Despite these general trends, beta diversity analyses revealed significant differences between communities even just a few meters apart, highlighting rapid spatial turnover, likely driven by patchy resource distribution and niche filtering [27,34].

A distance-decay relationship, wherein community similarity decreases with spatial separation, was evident, aligning with established patterns in soil fungal biogeography [35].

Dispersal limitation and localized environmental filtering appear to jointly shape fungal communities [36], even at fine spatial scales. Similar observations in ectomycorrhizal fungi and other arid soil microbiomes support this finding [37].

Several mechanisms may explain spatial changes in fungal composition around *Z. lotus.* The shrub’s roots improve water retention, release exudates that serve as microbial substrates, and trap litter, all contributing to a richer soil environment [38]. Our findings confirmed elevated levels of organic matter, nitrogen (N) and phosphorus (P) at 0 m [27], validating this shrub-mediated nutrient enrichment.

We also found that specific edaphic factors, CaCO_3_, NO_3_^−^, and Na_2_O, strongly influenced fungal composition. High CaCO_3_, indicative of alkaline conditions, is known to shape fungal communities [39]. Nitrate availability, as a key microbial nutrient, may favor copiotrophic fungi, whereas oligotrophs dominate in N-depleted zones. Interestingly, our study suggests moderate nitrogen levels near *Z. lotus* support a balanced and diverse community. High Na_2_O levels reflect salinity stress, which selectively favors salt-tolerant taxa such as Ascomycota [30]. However, it is important to note that our study did not directly measure litter accumulation, root biomass, or soil moisture; factors known to influence soil fungi. We infer their effects indirectly, but future studies should quantify these variables. Which will help disentangle the mechanisms behind the patterns we observed.

Shannon–Wiener and Simpson indices indicated that fungal diversity decreased with distance from *Z. lotus* (0 m > 3 m > 6 m). Root-proximal soils hosted enriched communities of multitrophic fungi such as *Pleiochaeta*, *Xenodidymella*, and *Aureobasidium*, many of which produce extracellular polymers for stress resistance [40]. Fast-growing saprotrophs (*Chaetomium*, *Penicillium*) thrived near organic matter-rich zones, while oligotrophs like *Alfaria* emerged at 6 m, consistent with resource gradient theory [41,42].

Interestingly, despite barley cultivation, fungal composition remained relatively stable around *Z. lotus*, possibly due to microbial buffering by root exudates or allelopathic effects. In contrast to studies where tillage disrupts fungal networks [43], we observed higher richness in cultivated plots, perhaps due to barley roots and inputs enhancing microbial niches. However, copiotrophic taxa such as *Fusarium* and *Alternaria* did not increase at 6 m under cultivation, suggesting inhibitory effects, potentially from allelochemicals derived from *Z. lotus* [44].

FUNGuild analyses confirmed functional redundancy in dominant genera. Many taxa shift among saprotrophic, symbiotic, and pathogenic modes, a strategy also seen in Mediterranean shrubs [45] For example, *Fusarium* can become pathogenic under host stress, and *Cladosporium* spores resist desiccation. Such conditional mutualisms, where opportunists are tolerated for their ecological services, may be essential for plant survival in semi-arid ecosystems [46].

Co-occurrence network analysis revealed key taxa under different land-use conditions. *Pleiochaeta* dominated planted fields, likely benefiting from root-associated substrates, while *Aspergillus* was central in non-planted soils, consistent with its saprotrophic lifestyle and dominance in undisturbed soils [47,48]. These patterns illustrate how cultivation shifts community composition toward plant-associated fungi, while non-planted soils support generalist decomposers.

## 5. Conclusions

This study demonstrates that *Ziziphus lotus* significantly influences fungal community structure in arid soils. Fungal richness and diversity were highest beneath the shrub canopy and declined with distance, with clear shifts in community composition associated with edaphic variables such as nitrate, salinity, and calcium carbonate. The shrub’s microenvironment supported fungi with diverse ecological roles, ranging from saprotrophs to multitrophic taxa, reflecting adaptations to arid and nutrient-limited conditions. Notably, when comparing land-use types, fungal diversity was generally higher in the planted fields than in the non-planted field, although overall community structure remained broadly similar. This indicates that *Z. lotus* maintains a relatively stable and resilient fungal community even under cultivation, underscoring its role as a keystone shrub that facilitates soil microbial diversity and may contribute to sustainable land management in drylands.

However, we caution that these findings are based on a single region and two adjacent fields. While the results offer important insights, they should not be overgeneralized to other ecosystems without further validation. Future studies incorporating broader geographic sampling, temporal replication, and functional assays will be essential to confirm and expand upon the patterns observed here.

## Figures and Tables

**Figure 1 microorganisms-13-02489-f001:**
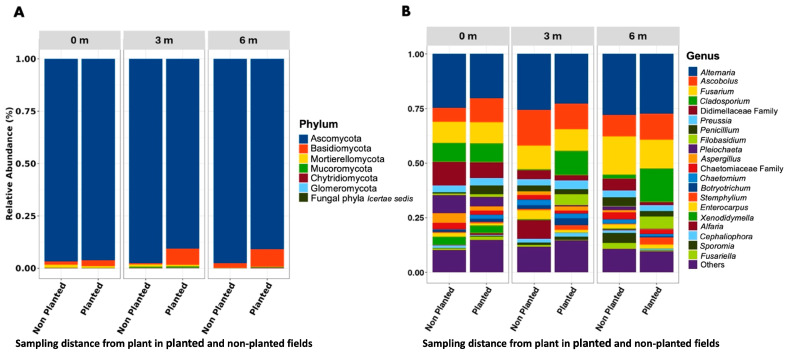
Distribution of the 20 most prevalent taxa, categorized by (**A**) phylum and (**B**) genus. Bars represent mean relative abundance by field type (non-planted vs. planted) within each distance class (0 m, 3 m, 6 m). Taxa not ranked among the top 20 are grouped under ‘Others’.

**Figure 2 microorganisms-13-02489-f002:**
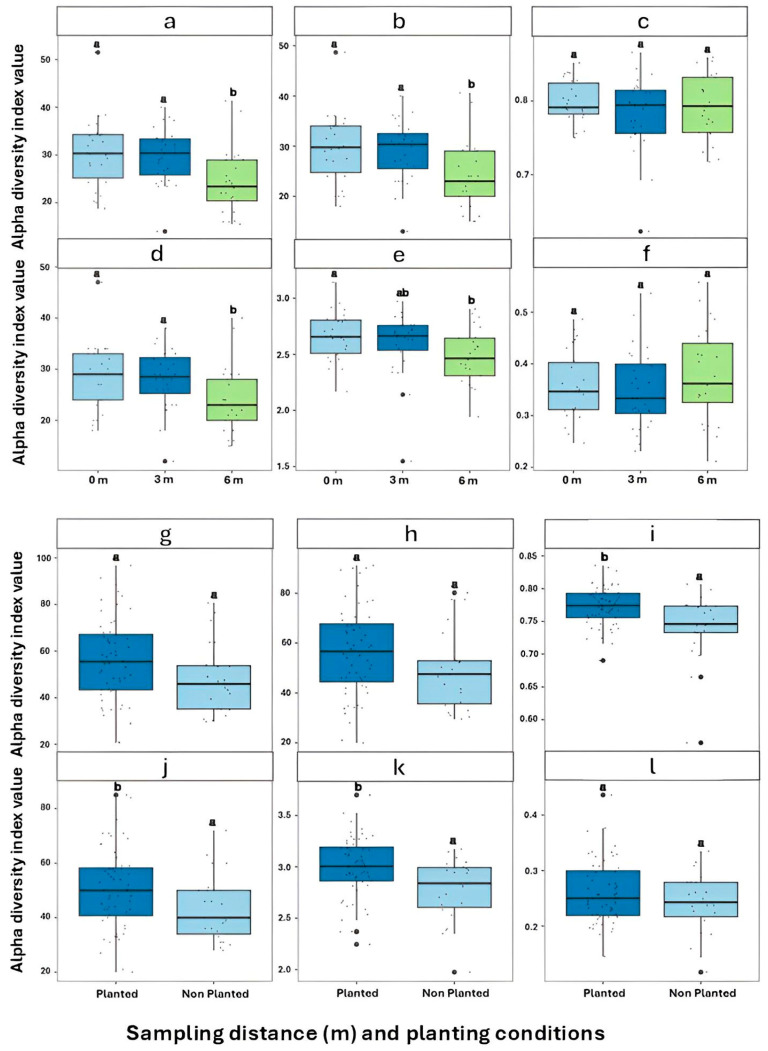
Alpha diversity indices of fungal communities across (**a**–**f**) sampling distance gradients (0 m, 3 m, 6 m) and (**g**–**l**) planting conditions (Planted vs. Non-planted) in Ziziphus lotus rhizosphere soils. (**a**,**g**) Observed richness; (**b**,**h**) Chao1; (**c**,**i**) Pielou’s evenness; (**d**,**j**) Shannon–Wiener index; (**e**,**k**) Inverse Simpson index; (**f**,**l**) Simpson’s evenness. Different letters (a, b) above boxplots indicate significant differences among groups.

**Figure 3 microorganisms-13-02489-f003:**
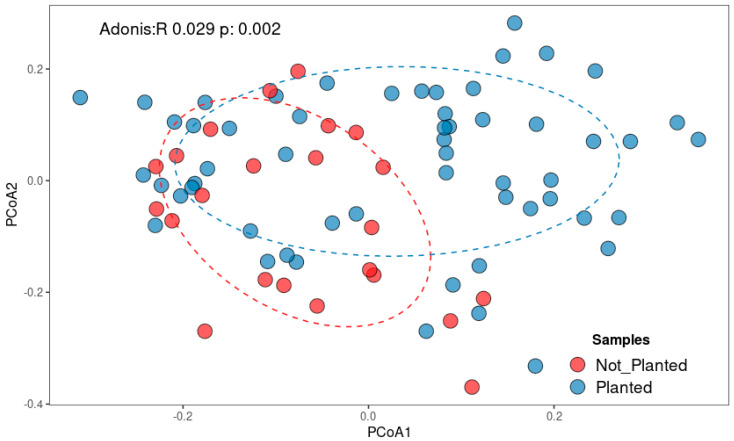
Spatial distribution patterns across the sample origins, visualized using principal coordinate analysis (PCoA) using the Bray–Curtis dissimilarities.

**Figure 4 microorganisms-13-02489-f004:**
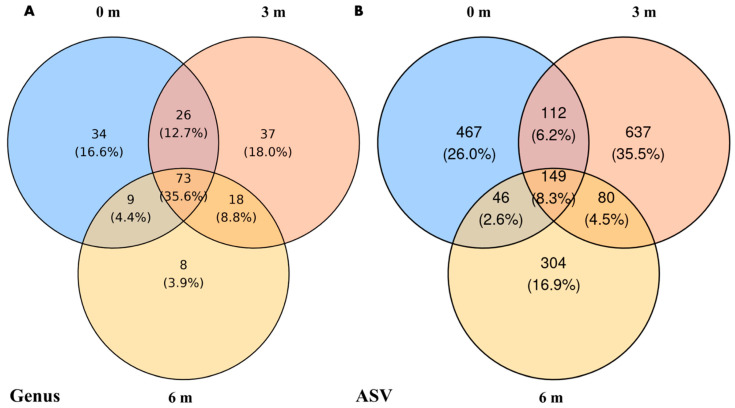
Venn diagrams showing the distribution of taxa across different sampling distances, categorized by (**A**) genus level and (**B**) ASVs.

**Figure 5 microorganisms-13-02489-f005:**
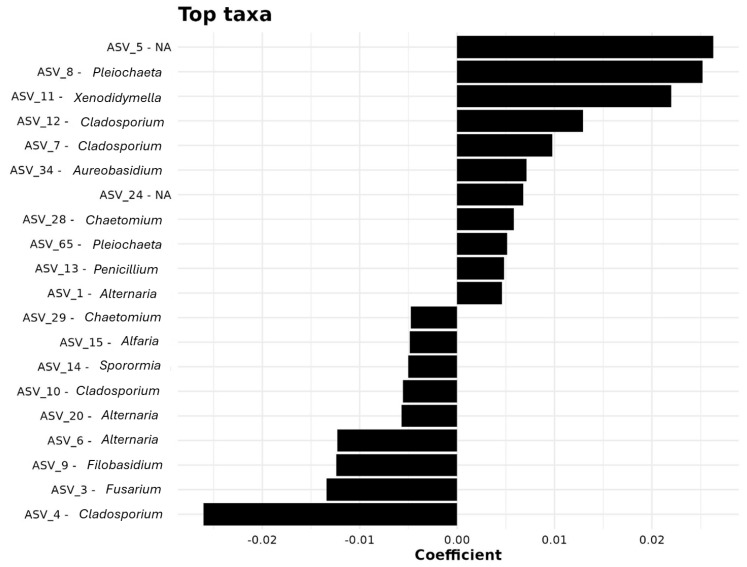
Key taxa influencing microbial community structure, identified through coefficient analysis.

**Figure 6 microorganisms-13-02489-f006:**
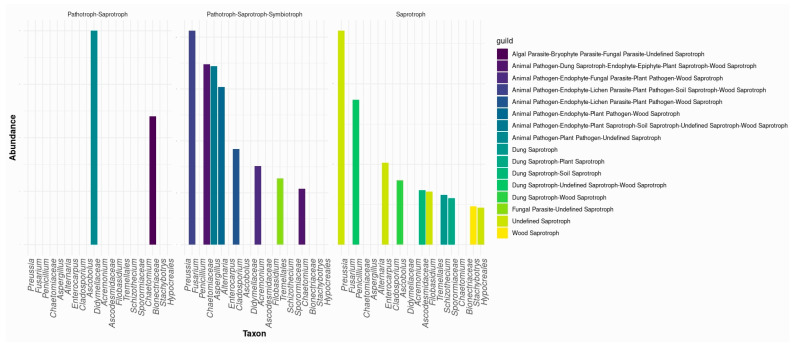
The top 20 fungal taxa and their predicted trophic modes and functions.

**Table 1 microorganisms-13-02489-t001:** Influence of selected chemical parameters on community structure at different distance levels (0 m, 3 m, and 6 m). Significance levels (*p*-values) are provided, with asterisks indicating statistically significant differences (* *p* ≤ 0.05, ** *p* ≤ 0.01, ns: Non significance).

Distance	Factor	*p*-Value	R2	Chi_Square_Percent	Significance
0 m	Total P	0.029	0.06	6.30	*
	Total K	0.004	0.07	7.09	**
	Total N	0.024	0.06	6.20	*
3 m	Total P	0.547	0.04	3.53	ns
	Total K	0.726	0.03	3.21	ns
	Total N	0.784	0.03	2.99	ns
6 m	Total P	0.263	0.05	4.69	ns
	Total K	0.506	0.04	3.88	ns
	Total N	0.195	0.05	5.05	ns

**Table 2 microorganisms-13-02489-t002:** PERMANOVA test of distance using the Bray–Curtis method. NA: not applicable.

Term	Df	Sum of Sqs	Mean Sqs	F. Model	R2	Pr (>F)
Distance in meters	2	1.512832	0.756416	3.516763	0.080735	0.001
Group	5	1.643269	0.328654	1.527992	0.087696	0.001
Distance in meters:Group	10	2.461678	0.246168	1.144495	0.131372	0.056
Residuals	61	13.12041	0.215089	NA	0.700196	NA
Total	78	18.73819	NA	NA	1	NA

**Table 3 microorganisms-13-02489-t003:** Factors significantly influencing alpha diversity, assessed by the ACE, Chao1, Pielou evenness, richness, Shannon–Wiener, and Simpson evenness indices, across different sampling distances and origins. Significance levels (*p*-values) are shown, with asterisks (* *p* ≤ 0.05, ** *p* ≤ 0.01, *** *p* ≤ 0.001, ns: not significant) indicating significant differences.

Index	Group1	Group2	*p*	*p*.Adj	*p*.Format	*p*.Signif	Method
	0 m	3 m	0.747188	0.75	0.74719	ns	T-test
Shannon–Wiener	0 m	6 m	0.001209	0.0024	0.00121	**	T-test
	3 m	6 m	0.000724	0.0022	0.00072	***	T-test
	0 m	3 m	0.688847	0.69	0.68885	ns	T-test
Richness	0 m	6 m	0.001406	0.0028	0.00141	**	T-test
	3 m	6 m	0.000208	0.00062	0.00021	***	T-test
	0 m	3 m	0.99059	0.99	0.99	ns	T-test
Pielou_evenness	0 m	6 m	0.123334	0.37	0.12	ns	T-test
	3 m	6 m	0.208024	0.42	0.21	ns	T-test
	0 m	3 m	0.727331	0.73	0.72733	ns	T-test
ACE	0 m	6 m	0.001937	0.0039	0.00194	**	T-test
	3 m	6 m	0.000193	0.00058	0.00019	***	T-test
	0 m	3 m	0.908342	1	0.91	ns	T-test
Simpson_evenness	0 m	6 m	0.340626	1	0.34	ns	T-test
	3 m	6 m	0.45281	1	0.45	ns	T-test
	0 m	3 m	0.520831	0.52	0.5208	ns	T-test
Chao1	0 m	6 m	0.00318	0.0064	0.0032	**	T-test
	3 m	6 m	0.0002	6 × 10^−4^	0.0002	***	T-test
Shannon–Wiener	Non-Planted	Planted	0.005116	0.0051	0.0051	**	T-test
Richness	Non-Planted	Planted	0.034111	0.034	0.034	*	T-test
Pielou_evenness	Non-Planted	Planted	0.011846	0.012	0.012	*	T-test
ACE	Non-Planted	Planted	0.039148	0.039	0.039	*	T-test
Simpson_evenness	Non-Planted	Planted	0.128035	0.13	0.13	ns	T-test
Chao1	Non-Planted	Planted	0.062234	0.062	0.062	ns	T-test

## Data Availability

The raw sequencing reads have been deposited in NCBI (https://www.ncbi.nlm.nih.gov/) with accession numbers provided under the BioProject ID: PRJNA1253578.

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
