# Peer review of "Spatial Structuring of Soil Fungal Diversity Associated with *Ziziphus lotus* (Rhamnaceae) in Arid Agricultural Soils"

_microorganisms, 2025, doi:10.3390/microorganisms13112489_

Round 1

Reviewer 1 Report

Comments and Suggestions for Authors

Dear Authors,
The study studied the "Spatial Structuring of Soil Fungal Diversity Associated with Ziziphus lotus (Rhamnaceae) in Arid Agricultural Soils. It is interesting, yet there were a large amounts of mistakes in the manuscript.

Specific comments:

  1. The article states that "its thorny branches can damage agricultural machinery and hinder field operations"; however, the significance of this characteristic in planted land remains unclear. Consequently, it is important to clarify whether the study aims to investigate the conservation of Ziziphus lotus in agricultural areas and its potential ecological functions.
  2. Shannon is commonly referred to as "Shannon-Wiener" in academic and technical literature.
  3. Lines 22~23, Soil physicochemical parameters how significantly influenced fungal diversity?
  4. 2. Line 28, Could you specify the different types of lifestyles and provide a comprehensive list?
  5. The terms "cultivated and fallow fields" and "planted and non-planted fields" are used inconsistently throughout the text. Please standardize their usage across the entire document.
  6. Lines 89–90, The land use history for both the planted and non-planted fields has not been provided.
  7. The article has conducted numerous statistics, but it has not detailed their specific methods.
  8. To ensure representativeness, it is necessary to combine the four sub-samples into a single composite sample. This research has been accomplished. However, the non-planted land used as a control group consists of only two replicates, which falls short of the minimum requirement of three replicates for valid statistical analysis. This limitation poses a significant risk to data reliability. Therefore, at least one additional sampling unit should be added to the non-planted land. Ideally, the number of replicates should be increased to four, matching the replication level in the planted land.
  9. Table S1 should present statistical data rather than individual values; Table3, S2, and S3 had many errors, such as "**p ≤ 0.01, *p ≤ 0.001", "P_value=0,439"... It should be "***p ≤ 0.001", "P_value=0.439"...
  10. Please introduce Random forest models and elaborate on how to use this method to calculate the importance values of each indicator.
  11. Table 1, all characters within each cell should be left-aligned at the top.
  12. Line 170, CaCO3 should be modified; "p" should be italicized, and the entire text should be revised.
  13. It appears that sodium oxide,organic matter, Magnesium oxide (MgO), calcium oxide, and others were not included in the analytical method you described.
  14. What are the differences in fungal communities and relative abundances between planted and unplanted lands? The article did not mention.
  15. Figure 3 is not formatted properly. It should have a border. And there should be clear scores between different groups.
  16. The differences at the genus level and among ASVs across different distances shown in Figure 3 are notnecessary. A more appropriate approach would be to present a comparison between the planted and unplanted areas.
  17. The article does not include Table 4 and presents only Figure 5. However, Figure 5 should encompass three distance gradients to meet the study requirements. A more appropriate approach would be to display the comparison between the planted and unplanted areas.
  18. In Section 3.5, which genera should be listed and what functions do they have? What is the proportion?
  19. Figure S3 should not be analyzed by combining different levels; instead, each level should be represented in a separate graph.
  20. In the second paragraph of the Discussion, is there any supporting data for the argument presented? The relevant data and corresponding statistical analyses should be provided.
  21. Lines 338 to 342: The claim that fungi enhance plant resilience requires stronger support from plant physiological and ecological indicators. Currently, the study lacks corresponding data to substantiate this assertion.The focus of discussion section should instead be on the intrinsic mechanisms by which plants influence fungal communities and diversity. Consequently, several revisions are necessary to better emphasize the key points.
  22. The conclusion section does not present the results or conclusions regarding the comparison between planted and unplanted areas. Furthermore, it includes indicators that were not addressed in this study, such as organic inputs, root exudates, and possible allelopathy. Speculation should not form the basis of the conclusions.

Author Response

Response to Reviewer#1, point by point:

Comment 1: The article states that "its thorny branches can damage agricultural machinery and hinder field operations"; however, the significance of this characteristic in planted land remains unclear. Consequently, it is important to clarify whether the study aims to investigate the conservation of Ziziphus lotus in agricultural areas and its potential ecological functions.

Response 1: We appreciate this observation. In the Introduction, we have clarified that although Z. lotusprovides important ecological functions, it also poses management challenges in agricultural fields. We note that while conservation is essential in natural habitats, integrated management strategies are required in agricultural contexts to balance ecological functions with productivity. This emphasizes that our study focuses on the ecological functions of Z. lotus in agroecosystems while acknowledging the need for careful management.

Comment 2: Shannon is commonly referred to as "Shannon-Wiener" in academic and technical literature.

Response 2: Thank you for the suggestion. In the revised manuscript, we have standardized the terminology and now refer to the index as Shannon–Wiener.

Comment 3: Lines 22~23, Soil physicochemical parameters how significantly influenced fungal diversity?

Response 3: We have clarified in the Abstract and Results how soil physicochemical parameters influenced fungal diversity, specifying which parameters showed significant effects and the nature of these effects.

Comment 4: Line 28, Could you specify the different types of lifestyles and provide a comprehensive list?

Response 4: We have revised the description of fungal lifestyles in both the Abstract and main text. Section 3.5 of the Results now provides a comprehensive list of the trophic modes identified by FUNGuild for the dominant taxa, giving a clear overview of the lifestyle categories.

Comment 5: The terms "cultivated and fallow fields" and "planted and non-planted fields" are used inconsistently throughout the text. Please standardize their usage across the entire document.

Response 5: We have standardized these terms throughout the manuscript.

Comment 6: Lines 89–90, The land use history for both the planted and non-planted fields has not been provided.

Response 6:  We clarified that, at the time of sampling, one field was actively cropped and managed as an agricultural field, while the other was not planted that season. This provides context for “planted” vs. “non-planted” sites.

Comment 7: The article has conducted numerous statistics, but it has not detailed their specific methods.

Response 7: We expanded the Statistical Analysis section to describe all diversity metrics, transformations, and tests. Libraries were rarefied, α-diversity indices calculated (Shannon–Wiener, inverse Simpson, Pielou’s evenness, Simpson’s evenness, observed richness, Chao1, ACE), and group differences assessed with ANOVA followed by multiple-comparison tests. β-diversity (Bray–Curtis) was analyzed using PCoA and PERMANOVA (adonis) in R with phyloseq, vegan, and supporting packages. Figure code and outputs are provided.

Comment 8: To ensure representativeness, it is necessary to combine the four sub-samples into a single composite sample. This research has been accomplished. However, the non-planted land used as a control group consists of only two replicates, which falls short of the minimum requirement of three replicates for valid statistical analysis. This limitation poses a significant risk to data reliability. Therefore, at least one additional sampling unit should be added to the non-planted land. Ideally, the number of replicates should be increased to four, matching the replication level in the planted land.

Response 8: We acknowledge this limitation. Each sample was a composite of four sub-samples to maximize representativeness. The non-planted field had only two Z. lotus shrub patches, and no additional patches were available.

Comment 9: Table S1 should present statistical data rather than individual values; Table3, S2, and S3 had many errors, such as "**p ≤ 0.01, *p ≤ 0.001", "P_value=0,439"... It should be "***p ≤ 0.001", "P_value=0.439"...

Response 9: We corrected formatting and notation in all tables. Table S1 retains raw physicochemical values for transparency; p-values now follow conventional formatting (italic lowercase p, decimals with a period, asterisks for significance: *p ≤ 0.05, **p ≤ 0.01, ***p ≤ 0.001).

Comment 10: Please introduce Random forest models and elaborate on how to use this method to calculate the importance values of each indicator.

Response 10: We added a clear description of Random Forest models in the Methods section, explaining model construction and calculation of taxon importance values.

Comment 11: Table 1, all characters within each cell should be left-aligned at the top.

Response 11: Table 1 cells are now left- and top-aligned.

Comment 12: Line 170, CaCO3 should be modified; "p" should be italicized, and the entire text should be revised.

Response 12: These formatting and methodological clarifications were implemented throughout the manuscript.

Comment 13: It appears that sodium oxide,organic matter, Magnesium oxide (MgO), calcium oxide, and others were not included in the analytical method you described.

Response 13: These methodological clarifications were implemented throughout the manuscript.

Comment 14: We explicitly describe differences in the Results and Discussion: certain taxa (e.g., Cystofilobasidiaceae) were enriched in planted soils, while Stephanosporaceae, Myriococcum, and Alfariawere more abundant in non-planted soils. Shannon–Wiener diversity and other indices were higher in the planted field (p-values 0.0051–0.04).

Comment 15: Figure 3 is not formatted properly. It should have a border. And there should be clear scores between different groups.

Response 15: The formatting of Figure 3 was implemented.

Comment 16: The differences at the genus level and among ASVs across different distances shown in Figure 3 are notnecessary. A more appropriate approach would be to present a comparison between the planted and unplanted areas.

Response 16: Modifications were made accordingly.

Comment 17: The article does not include Table 4 and presents only Figure 5. However, Figure 5 should encompass three distance gradients to meet the study requirements. A more appropriate approach would be to display the comparison between the planted and unplanted areas.

Response 17: We have now added Table 4 to the Supplementary Materials (previously omitted by oversight), which lists the top indicator taxa and their importance values. Figure 5 in the revised manuscript directly corresponds to these data, showing the taxa that were most influential across the three distance gradients (0, 3, and 6 m). To address the reviewer’s suggestion regarding comparisons between planted and non-planted fields, the Figure S3 present the results of the differential abundance analysis showing the taxa that were enriched planted vs non planted field.

Comment 18: In Section 3.5, which genera should be listed and what functions do they have? What is the proportion?

Respond 18: We agree and have expanded Section 3.5 to name the principal genera, state their predicted trophic modes. Several modifications were added to address the reviewer’s comment, please to see the revised version of the MS.

Comment 19: Figure S3 should not be analyzed by combining different levels; instead, each level should be represented in a separate graph.

Response 19: We have replaced the original figure with a new version that specifically highlights the differences in fungal taxa between planted and non-planted fields. The updated figure presents the mean relative proportions of each taxon in both conditions, along with 95% confidence intervals and corrected p-values. This approach provides a clearer and statistically more robust representation of the compositional differences than the original figure, directly addressing the reviewer’s concern.

Comment 20: In the second paragraph of the Discussion, is there any supporting data for the argument presented? The relevant data and corresponding statistical analyses should be provided.

Response 20: The second paragraph of our Discussion originally contained a general argument about Z. lotuscreating fertile microhabitats that enhance fungal diversity, but we had not explicitly tied it to our own data in that draft. We have now strengthened this part of the Discussion by directly incorporating supporting data from our study.

Comment 21: Lines 338 to 342: The claim that fungi enhance plant resilience requires stronger support from plant physiological and ecological indicators. Currently, the study lacks corresponding data to substantiate this assertion. The focus of discussion section should instead be on the intrinsic mechanisms by which plants influence fungal communities and diversity. Consequently, several revisions are necessary to better emphasize the key points.

Response 21: We have revised the Discussion to remove speculative claims about fungi directly enhancing Z. lotus resilience, as our study did not measure plant physiology. Instead, we emphasize the mechanisms by which Z. lotus shapes fungal diversity, which is what our data support.

Comment 22: The conclusion section does not present the results or conclusions regarding the comparison between planted and unplanted areas. Furthermore, it includes indicators that were not addressed in this study, such as organic inputs, root exudates, and possible allelopathy. Speculation should not form the basis of the conclusions.

Response 22: We have revised the conclusion based on remarks addressed by the reviewer.

Reviewer 2 Report

Comments and Suggestions for Authors

The authors have submitted an article titled "Spatial Structuring of Soil Fungal Diversity Associated with Ziziphus lotus (Rhamnaceae) in Arid Agricultural Soils." This paper falls within the scope of the journal "Microorganisms". However, after critically reviewing the paper, I found several things that need to be changed or added before this paper can be published. Below you will find my (brief) comments on this:

  • The description of the experiment (lines 78-82) has no place in the introduction.
  • It would be helpful for the reader if the objectives at the end of the introduction were numbered (I, II, etc.).
  • When describing the meteorological conditions, the average annual temperature and rainfall distribution must also be included. Furthermore, the source of the meteorological data must be clearly stated.
  • I still don't understand why the sampling design requires that 5 shrub patches be sampled in the barley field, while only 2 are sampled in the fallow areas. This needs to be explained in more detail.
  • Even though the methods used to determine certain soil physical and chemical parameters have already been described in a previous paper, both the methodology used and the analytical equipment (manufacturer, type) should be mentioned here. I would also like to add that Table S1 presents results for many more parameters than are listed in the methodology section.
  • 170: The 3 in CaCO3 needs to be a subscript.
  • Table 1 does not show "selected environmental variables" but rather selected soil chemical parameters.
  • The x-axis labels are missing in Figure 1.
  • Figure 2 is missing all axis labels. Furthermore, I recommend that the subplots (ACE, Chao1, Shannon, etc.) not be labeled with their full names, but rather abbreviated with letters (a, b, c, etc.), with the full names then provided in the figure caption. For clarity, I also recommend splitting Figure 2 into two separate figures (one for alpha diversity and one for planted vs. non-planted areas).
  • In Figure 5, one of the two sub-graphs should be displayed in a different color for better clarity. The results shown in Figure 5 (lines 269-272) should be described in much greater detail.
  • The map source for Figure S1 must be mentioned in the figure caption. In addition, a north arrow must be added. The date of the images in Figure S1B and S1C must also be included.

Author Response

Response to Reviewer#2, point by point:

Comment 1: The description of the experiment (lines 78-82) has no place in the introduction.

Response 1: The description of the experiment was removed from the introduction.

Comment 2: It would be helpful for the reader if the objectives at the end of the introduction were numbered (I, II, etc.).

Response 2: Introduction was revised, with objectives now numbered i, ii, and iii.

Comment 3: When describing the meteorological conditions, the average annual temperature and rainfall distribution must also be included. Furthermore, the source of the meteorological data must be clearly stated.

Response 3: We added average annual temperature, rainfall distribution, and data sources.

Comment 4: I still don't understand why the sampling design requires that 5 shrub patches be sampled in the barley field, while only 2 are sampled in the fallow areas. This needs to be explained in more detail.

Response 4: Please see our reply to Reviewer 1, Comment 8. The discrepancy in the number of patches, 5 in the planted field versus 2 in the non-planted field, was due to the characteristics of the study site: only two Z. lotus shrub patches were available in the non-planted field, while the planted field contained five accessible patches.

Comment 5: Even though the methods used to determine certain soil physical and chemical parameters have already been described in a previous paper, both the methodology used and the analytical equipment (manufacturer, type) should be mentioned here. I would also like to add that Table S1 presents results for many more parameters than are listed in the methodology section.

Response 5: Methods updated with equipment details; tables revised and formatting corrected.

Comment 6: 170: The 3 in CaCO3 needs to be a subscript.

Response 6: Corrected.

Comment 7: Table 1 does not show "selected environmental variables" but rather selected soil chemical parameters.

Response 7: Changed.

Comment 8: The x-axis labels are missing in Figure 1.

Response 8: Figure 1 revised with axis labels.

Comment 9: Figure 2 is missing all axis labels. Furthermore, I recommend that the subplots (ACE, Chao1, Shannon, etc.) not be labeled with their full names, but rather abbreviated with letters (a, b, c, etc.), with the full names then provided in the figure caption. For clarity, I also recommend splitting Figure 2 into two separate figures (one for alpha diversity and one for planted vs. non-planted areas).

Response 9: Figure 2 revised with axis labels, panel letters, color adjustments, and improved clarity. Figure 2 remains combined for conciseness.

Comment 10: In Figure 5, one of the two sub-graphs should be displayed in a different color for better clarity. The results shown in Figure 5 (lines 269-272) should be described in much greater detail.

Response 10: We have addressed the comment based on earlier remark by Reviewer 1. We have changed the figure to a more representative data with the same information to address both comments.

Comment 11: The map source for Figure S1 must be mentioned in the figure caption. In addition, a north arrow must be added. The date of the images in Figure S1B and S1C must also be included.

Response 11: Caption updated with source, north arrow, and image dates included.

Reviewer 3 Report

Comments and Suggestions for Authors

 Ziziphus lotus (L.), a resilient shrub native to Moroccan’s arid regions, play an essential role to retain soil moisture and accumulate organic matter.

The research study aimed to investigate the spatial distribution of fungal communities associated with Z. lotus in planted and non-planted fields.

The authors emphasized that fungal richness and diversity were highest near shrub.

The article is generally well structured. The results obtained are in accordance with the established objectives and are interpreted statistically.

The bibliographical sources are relevant to this study, and the conclusions are in accordance with the purpose of the paper.

The study presents the theoretical and practical importance for the rational utilization of arid agricultural soils.

Author Response

Response to Reviewer#3

Comment 1: Ziziphus lotus (L.), a resilient shrub native to Moroccan’s arid regions, play an essential role to retain soil moisture and accumulate organic matter.

The research study aimed to investigate the spatial distribution of fungal communities associated with Z. lotus in planted and non-planted fields.

The authors emphasized that fungal richness and diversity were highest near shrub.

The article is generally well structured. The results obtained are in accordance with the established objectives and are interpreted statistically.

The bibliographical sources are relevant to this study, and the conclusions are in accordance with the purpose of the paper.

The study presents the theoretical and practical importance for the rational utilization of arid agricultural soils.

Response 1: We thank Reviewer 3 for their thorough reading and positive feedback. We appreciate the acknowledgment of the manuscript’s structure, appropriate statistical analyses, relevant references, and the practical significance of our findings.

Round 2

Reviewer 2 Report

Comments and Suggestions for Authors

Dear authors,

you have followed my instructions very carefully, with only two minor exceptions. Please add the label for the x-axis to Figure 1, and both the x-axis and y-axis labels to Figure 2. In my opinion, the editors can make these changes, so the manuscript doesn't need to be sent back to me.

Best regards

Author Response

Comment : you have followed my instructions very carefully, with only two minor exceptions. Please add the label for the x-axis to Figure 1, and both the x-axis and y-axis labels to Figure 2. In my opinion, the editors can make these changes, so the manuscript doesn't need to be sent back to me.

Response: Thank you very much for your positive feedback and for carefully reviewing our revision. We have now added the requested x-axis label to Figure 1, as well as both the x-axis and y-axis labels to Figure 2.